# Passive Transonic Shock Control on Bump Flow for Wing Buffet Suppression

## Davide Di Pasquale * and Simon Prince

SATM, Centre for Aeronautics, Cranfield University, College Road, Cranfield, Bedfordshire MK43 0AL, UK;
simon.prince@cranfield.ac.uk
* Correspondence: davide.dipasquale@cranfield.ac.uk

**Abstract:** Since modern transport aircraft cruise at transonic speeds, shock buffet alleviation is one indispensable challenge that civil transport research needs to be addressed. Indeed, in the transonic flow regime shock-induced separation and transonic buffet compromise the flight envelope of an aircraft, and therefore its operational safety and structural integrity. One possible solution is to control and delay the boundary layer separation. The aim of this work was to study whether sub-boundary layer scale period roughness, which locally increases the boundary layer displacement thickness, can act as a virtual shock bump, with aim of bifurcating the foot of the shock wave to reduce the shock's adverse effect on the boundary layer in the same way as solid shock bumps are known to act. This passive approach can then enhance the buffet margin, consequently extending the safe flight envelope. An experimental investigation was performed, applying this passive technique on a wind tunnel wall bump model which simulated the flow over the upper surface of an aerofoil. The results, in terms of surface pressure distribution and corresponding shadowgraph flow visualisation, showed that such periodic roughness can, indeed, bifurcate the shock wave and delay shock-induced separations, depending on the orientation of the roughness and its periodicity. A virtual shock bump effect can be produced using the displacement effect of periodic sub-boundary layer scale roughness.

**Keywords:** transonic aerodynamics; shock wave boundary layer interaction; passive flow control; surface roughness; buffet alleviation; experiment





## 1. Introduction

### 1.1. Background

The environmental impact is the primary driving force for cleaner, and quieter aircraft of the future. In order to respect the requirements of the International Air Transport Association (IATA) and governments on the reduction in $CO_2$ emissions from air transport, aircraft manufacturers have to improve further their aircraft performance and fuel system. A cost-effective way to reduce drag and, thereby, fuel consumption is to improve the aerodynamic efficiency. Since the advent of the jet engine, virtually all commercial transport aircraft now cruise in the transonic speed range, because this allows the range of the aircraft to be maximised. Transonic cruise is the longest mission phase of a civil airliner, in which the majority of the fuel is consumed. A mixture of supersonic and subsonic flow characterizes the transonic regime. The transonic flow regime has been, and remains, a challenge both for computational prediction and experimental simulation. The close coupling of the shock waves arising from the compressibility of the air and the viscous flow on the aircraft surfaces leads to highly unsteady and complex flows that often involve detrimental flow separations. The general design concept for a critically designed aircraft wing allows for an extended supersonic region over the wing upper surface while having a weak normal shock separating it from the subsonic flow. Beyond a critical Mach number, shock waves appear in the flow field, becoming stronger as the speed increases.

The separation of the boundary layer depends mainly on the adverse pressure gradient imposed by the shock wave and the subsequent pressure recovery to the wind trailing edge. The boundary layer tends to dissipate the sharp pressure rise across a weak shock wave and can remain attached to the surface if the shock is not too strong.

In Figure 1, the supersonic region ($M > 1$) of the transonic flow around an aerofoil is delimited by a sonic line ($M = 1$). The size of this region depends on the velocity of the flow in the free stream. The location of the shock and its strength depend indirectly on the size of this supersonic region. After the shock, the flow becomes subsonic again ($M < 1$), and the deceleration causes a strong adverse pressure gradient which is transmitted through the boundary layer. Within the boundary layer, there is a sonic line separating the supersonic and subsonic regions. Below this line, downstream pressure information can travel backwards and propagate upstream. This affects the strength, shape and position of the shock. This "upstream influence" results in compression waves being generated at the foot of the shock wave, dissipating it near the wall, thus producing a decrease in stagnation pressure losses.

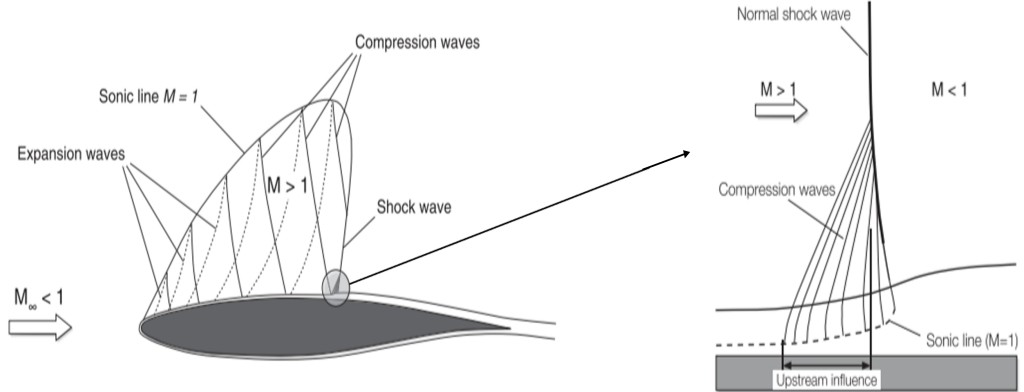

**Figure 1.** A typical shock wave–boundary layer interaction (SBLI) on a transonic wing [1].

When the boundary layer is laminar, it almost always detaches from the surface. In practice, however, a turbulent boundary layer is most often observed, which can withstand large pressure increases before becoming detached. The strength of the shock depends, among other things, on the speed of the flow. When a so-called 'strong' shock is generated, the boundary layer can separate from the surface, severely altering the flow conditions. If the upstream boundary layer is laminar a local shock-induced separation can occur followed by a reattachment to the surface a little further downstream forming a bubble. A complete detachment from the surface occurs from the shock to the trailing edge tends to occur for fully turbulent flow. The latter is called "shock stall" due to a sudden loss of lift and a considerable increase in drag. A weak shock also tends to increase the thickness of an attached boundary layer, which increases viscous losses.

The shock waves lead to a rapid increase in drag, both due to the rise in wave drag, due to the loss of total pressure through the shock, and also because the pressure rise through a shock wave thickens the boundary layer, leading to increased viscous drag. Transonic shock buffet is a phenomenon where an interaction between shock waves and the separated shear-layer leads to a self-sustained periodic shock motion that in turn involves large fluctuations of wing loading [2]. From passenger comfort to potential wing structural failure, the buffet develops gradually over a wing as its incidence increases [3,4]. The buffet phenomenon is therefore a performance limiting factor since it constrains the flight envelope of an aircraft calling for airworthiness regulations to set requirements on a sufficient margin before onset, in both flight Mach number and angle of attack. Safety is the single most important principle in aviation, and ACARE has announced the aim for having less than one accident for each ten million commercial flights in Europe. This means that buffet suppression technology is of critical importance. Numerous research studies

have been undertaken to investigate shock wave–boundary layer interactions (SWBLI) with a view to identify ways to suppress the shock buffet in order to extend the flight envelope. Several flow control techniques, both active and passive, have been studied in the literature for controlling the SWBLI, mainly aimed at improving the off-design performance (i.e., delaying buffet onset and increasing drag-divergence Mach number). This, however, might often come at the expense of slightly increasing the overall drag force on the aircraft. Controlling the interactions between shock waves and boundary layers is the key principle behind most buffet control methods.

The active control methods such as the trailing-edge deflector [5,6], trailing edge flap [7], and fluidic vortex generator [8] were discovered to be the most efficient methods for controlling buffet under a variety of flow conditions, but they require extra energy and more complex systems to be applied which can add a lot of extra mass. Due to their simplicity in comparison to the active approaches, the passive solutions for buffet alleviation, such as the stream-wise slots [9,10], the shock control bumps (SCBs) [11,12], and the vortex generators (VGs) [13,14], were also investigated. Typical vortex generators and synthetic jets are only one of the three categories discussed by [15] for buffet control. The idea is to create a boundary layer that is less susceptible to adverse pressure gradients by energizing the flow upstream of a shock wave. Although VGs have been found useful to delay buffet onset at higher incidence angles, drag rise at cruise is a major drawback of this category [16] of approach. The use of shock control bumps has been marked as a beneficial technique, although research is still ongoing, to assess the feasibility given the possible drag liability. However, on the practical side of things, applying bumps to a wing is expensive since they will add mass and involve a redesign of the wing if integrated properly [17]. Additionally, the bumps are not easily applicable. This is because of the flexing nature of the wing at different flight conditions and manoeuvres.

### 1.2. Shock Control Using Distributed Roughness

A new passive buffet suppression technique suggested in this paper, uses of sub-boundary layer scale roughness arranged in periodic strips such that their local effect in increasing boundary layer displacement thickness can act as virtual shock bumps. If these periodic strips were also to be skewed at an angle to the oncoming flow, in a similar way as vortex generators are, the idea would be to also induce increased vorticity into the boundary layer, thereby enhancing the mixing and re-energisation effect. As with solid shock bumps the aim would be to bifurcate the transonic shock wave into two weak oblique shocks at the foot of the shock in order to develop a $\lambda$ shock structure. This will result in reducing the total pressure loss across the shock system compared with the case of a single stronger shock wave. This will also result in a more benign adverse pressure gradient which may suppress boundary layer separation under the shock. Very little research has been conducted on the effect of distributed roughness configurations on SWBLIs.

One study, by Dietz [18], looked at a novel approach by applying D-strips (similar to surface patches) on the surface of the VC-Opt supercritical aerofoil. These strips were Velcro-type steps that were placed underneath the shock wave location. Results from their experiment showed that the D-strips were able to successfully weaken the shock by bifurcation, thereby reducing the prominent effect of wave drag. However, the paper concluded that the total drag registered by the wake rake, directly downstream of the strips, showed a higher measurable value compared to the span-wise locations away from the strips where a stronger shock wave was present. Nevertheless, the overall result showed a positive delay in the buffet onset. This was seen with the shock wave moving downstream compared to the baseline case with no control. The effect of surface roughness on the SWBLI in transonic flows has also been investigated in the literature by Babinsky at al. [19].

### 1.3. Roughness Effect

The role of surface roughness in many flow applications is of great practical importance as a well-known effect is the acceleration of the laminar–turbulent transition process. The

implications of the premature appearance of turbulent flow are considerable due to changes in the aerodynamic loading, heat transfer, and gas mixing properties that result from transition.Roughness can be characterised as a surface with asperities (e.g., grains of sand). This is defined as the deviation of the real surface from the ideal smooth form in the direction of the normal vector. If this deviation is large, the surface is rough, whereas if it is small, the surface is aerodynamically smooth. The roughness is characterized by its location, density and height compared with the local boundary layer height.

A rough surface, when placed in a flow, influences its behaviour. Past experiments have shown that surface roughness exerts a resistance on the flow, resulting in drag increase. Increased surface roughness is recognised to have a thickening effect on the boundary layer, increasing the local displacement thickness over and above that for a smooth surface. Another well-known effect of surface roughness is the change in velocity distribution and turbulence near the wall. This phenomenon has been quantified by experiments conducted on rough pipes and channels. Patel [20] was able to show that the near-wall velocity distribution on a rough surface had the same shape as on a smooth surface with a shift depending on the size of the roughness, following Equation (1).

$$u^+ = \frac{1}{k} \ln(y^+) + 5.5 - \Delta B \tag{1}$$

where $k$ is the von Kármán constant ($k = 0.41$) and $\Delta B$ is the function representing the offset due to the size and shape of the roughness, as provided in Equation (2).

$$\Delta B = \frac{1}{k} \ln(1 + 0.3k_s^+) \tag{2}$$

where

$$u^+ = u/u_\tau, \quad y^+ = u_\tau y/\nu, \quad k_s^+ = k_s u_t/\nu, \quad u_\tau = \sqrt{\tau_\omega/\rho} \tag{3}$$

Here, $u$ is the velocity at a distance $y$ from the wall, $k_s$ is the equivalent sand grain roughness height, $\nu$ is the kinematic viscosity of the fluid, and $u_\tau$ is the friction velocity defined by the wall shear stress $\tau_\omega$ and the fluid density $\rho$. In other words, the roughness modifies the law-of-the-wall used for the viscous sublayer.

It is possible to classify the flow into three regimes depending on the value of the dimensionless number $k_s^+$:

- $k_s^+ < 5$: The roughness is negligible in the smooth hydraulic regime;
- $5 < k_s^+ < 60$: The roughness becomes increasingly important in the transient regime;
- $k_s^+ > 60$: The roughness takes full effect in the fully rough regime.

The roughness effects are negligible in the hydrodynamically smooth regime, but become increasingly important in the transitional regime, and take full effect in the fully rough regime.

## 2. Methodology

The aim of this study was to assess the viability of periodic strips of sub-boundary layer scale-distributed roughness to control the transonic shock wave on the upper surface of a wing. A wall-mounted bump model, akin to the upper surface profile of an aerofoil was tested in transonic flow, where a near-normal shock wave, as shown in Figure 2, was generated that was capable of separating the wall boundary layer. Specifically, Figure 2 shows an example of a shock wave over the bump at a specific blowing pressure (35 psi) for the smooth case. For higher pressure the shock extends to the roof.

This baseline flow was then used as the basis of a study to investigate the effect of roughness configurations placed on the surface under the shock wave.

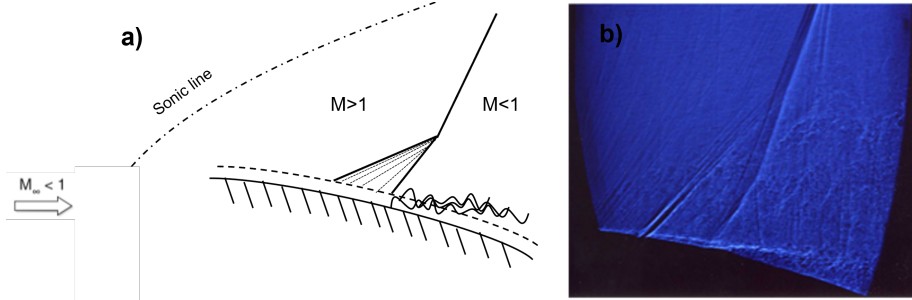

**Figure 2.** Shock wave over the bump. (**a**) Schematic; (**b**) shadowgraph image.

*The Experimental Setup*

Experiments were conducted in the City University of London T5 Transonic closed return Wind Tunnel, shown in Figure 3, which has a rectangular test section size of 0.2 m (height) × 0.255 m (width) × 0.5 m (length), and a Mach number range of 0.5–2. The working section was equipped with side windows with glass to allow shadowgraph flow visualisation to capture high-speed video images of the unsteady flow over the bump. The tunnel operated via an ejector system blowing four jets of air into the diffuser behind the working section, with the jet blowing pressures used to set the pressure ratio between the working section inlet (constant total pressure of 1 bar) and outlet.

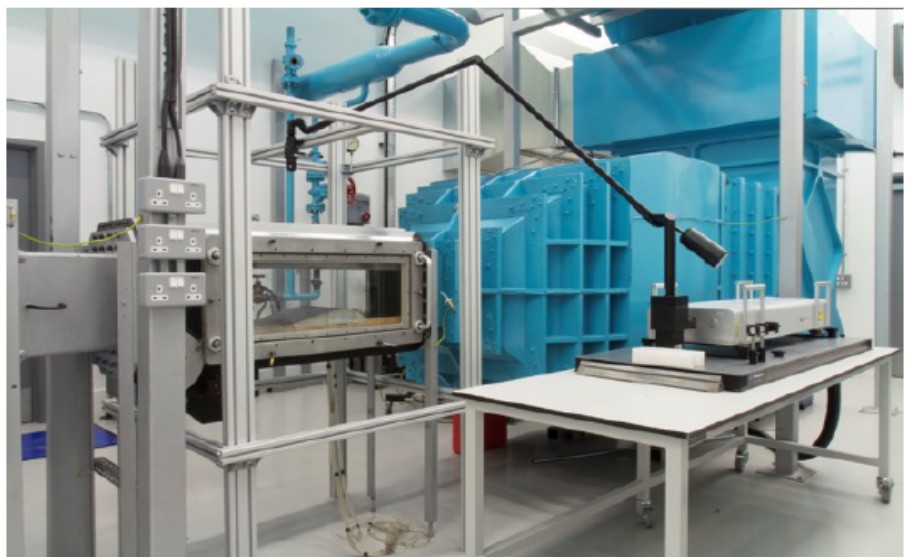

**Figure 3.** Transonic Wind Tunnel of the City University of London.

The bump with 14% $t/c$ ratio was placed on the test section and several configurations of distributed surface roughness configurations were tested. Ten millimetre roughness strips were placed on the top of a removable block embedded into the bump model, whose top surface followed the bump profile. The bump had a chord length of 309.55 mm with a maximum height of 43 mm. The roughness strips were placed between $x$ = 94 mm and $x$ = 147.5 mm, which defined the front and rear edge of the removable block. Figure 4 shows the bump geometry and roughness strip configurations used for the experiment. Pressure tappings were integrated into the bump model on the centreline of the model span ahead of and in rear of the removable block, on which there were no pressure measurement stations.

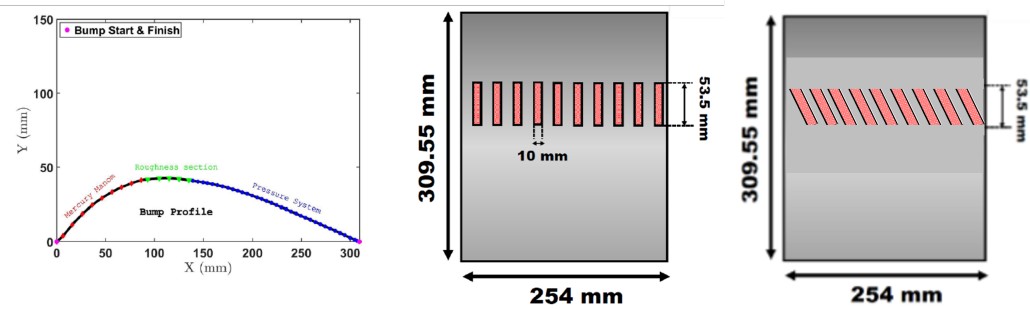

**Figure 4.** Bump geometry and roughness strip configurations.

The spacing between the strips as well as the angle of the strips varied according to the configurations detailed in Table 1.

Here, $\lambda$ is the width of the roughness strips, $\phi$ is the skew angle of the strips relative to the stream-wise/$x$-direction, $d$ is the spacing between the roughness strips and $n$ is the number of full strips which spanned the model in each case.

**Table 1.** Details of the roughness strips used in the experiment.

| $\phi$ (°) | $\lambda$ (mm) | k (μm) | d (mm) | n |
| --- | --- | --- | --- | --- |
| 0 | 10 | 500 | 5 | 16.67 |
| 0 | 10 | 500 | 20 | 8.33 |
| 30 | 10 | 500 | 5 | 16.67 |
| 30 | 10 | 500 | 10 | 12.5 |

The experiments with a smooth surface were also carried out in order to compare with the rough cases, and thus derive an understanding on the effects of the roughness configurations. The flow Mach number ahead of the bump was a constant value of $M = 0.55$, the flow being choked over the crest of the bump itself. To obtain the pressure distribution on the bump, Kulite pressure sensors were placed on the centreline axis along the surface behind the block insert, where the surface pressures would change with the tunnel pressure ratio. The typical static error band for the Kulite sensors used is less than 0.1% of the full scale. However, not enough of these were available for all of the tappings, and so a mercury manometer was connected to the few tappings ahead of the block insert, where the pressure would not change. The accuracy of these was estimated to be $\pm$1.37%, taking account the atmospheric pressure changes and barometer readings. Compressed air at 600 psi was stored in a reservoir out of which air was extracted at varying pressures to drive the tunnel via the ejector. A valve was manually operated to control the ejector jet pressure while the tunnel was in operation. Pressure readings and shadowgraph imagery was acquired 2 s after the manometer pressures were observed to stabilize.

The shadowgraph optical flow imaging system employed a standard vertical single knife edge assembly with an LED light source with the light path directed parallel to the working section windows by a 6 inch parabolic mirror, and focussed to a high-speed digital camera via a convex lens. The imagery was focused on the rear of the model where the shock system and separated flows were expected to be observed.

Kulite pressure data were acquired at 3 kHz over the last 5 s of the 15 s wind tunnel running. These data were then averaged over time to obtain a single value for each pressure tap. This procedure was applied for each configuration with eight different ejector blowing pressure settings (15, 20, 25, 30, 35, 40, 45 and 50 psi). The relationship between the ejector blowing pressure and tunnel outlet static pressure is shown in Figure 5, which presents a linear relation between the two.

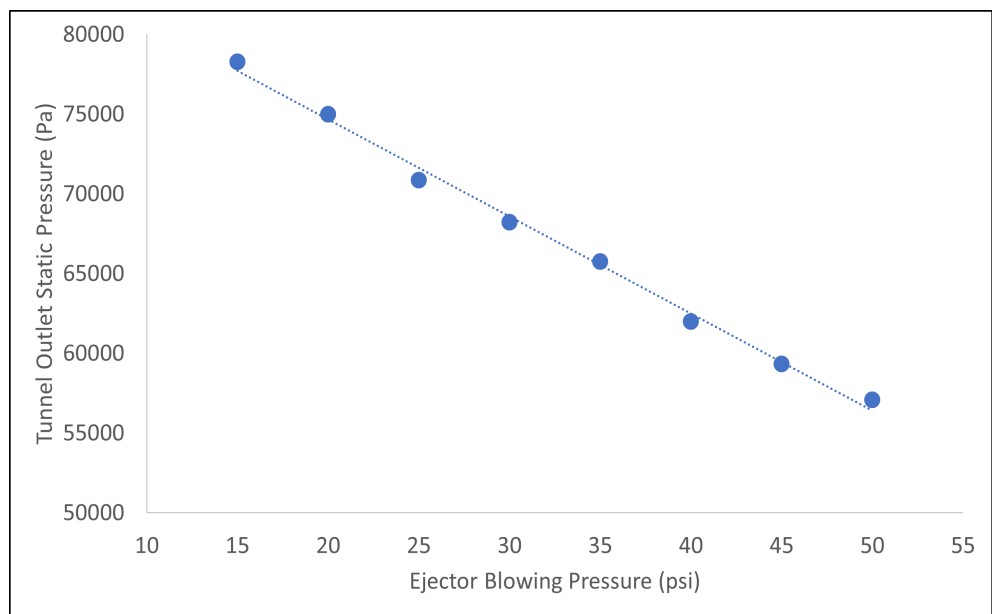

**Figure 5.** Tunnel outlet static pressure for the different ejector blowing pressures.

At least two sets of measurements were made for each blowing pressure to check the repeatability. Figure 6 shows an example of repeatability checks performed for two different blowing pressure.

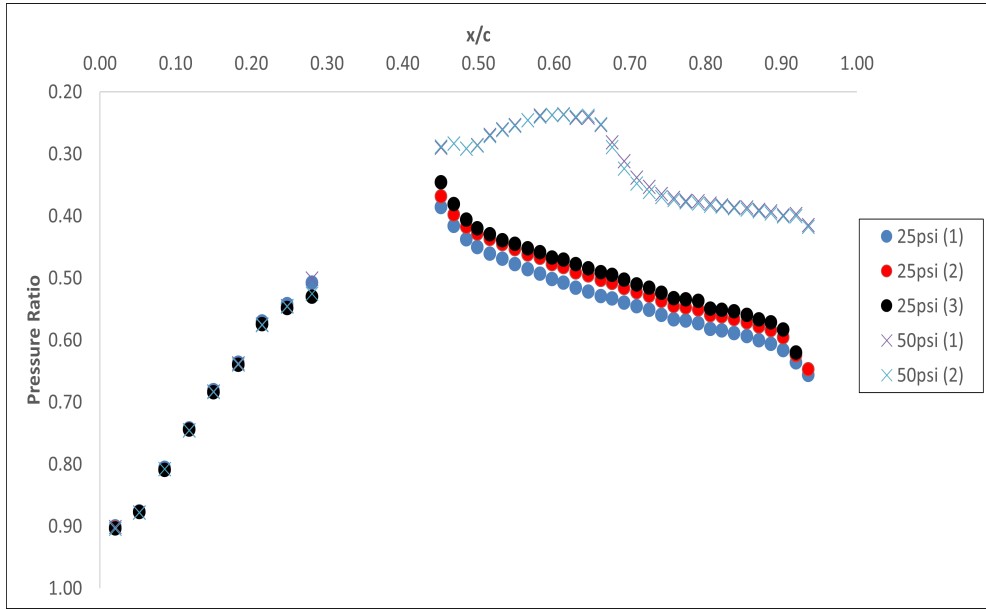

**Figure 6.** Repeatability check for two different ejector blowing pressures.

In Figure 7, the experimental results on the baseline smooth bump are presented in terms of the pressure ratio (local static pressure over the total pressure).

The pressure distribution up to an $x/c$ of 0.28 is nearly the same for all tested blowing pressures because this is in the subsonic region ahead of the supersonic flow over the crest of the bump, which chokes the flow. The flow returns to subsonic via a shock system on rear of the bump. The shock wave causes the boundary layer behind the shock to separate for large ejector blowing pressure cases, which in turn interacts with the shock wave by pushing it forward. Different pressure distributions are experienced at the upper and rear part of the bump. As the blowing pressure in the wind tunnel increases, a shock wave appears on the bump and becomes stronger. Between 30 and 35 psi ejector pressure the

shock wave is not strong enough to cause boundary layer separation; however, a sharp increase in $Cp$ caused by the shock wave is evident. The shock wave can be seen travelling aft with increasing ejector pressure as expected. For 30 psi the shock wave occurs at $x/c = 0.47$ and for 50 psi at $x/c = 0.63$, and the length of the interaction with the boundary layer increases. At the rear part of the bump the beginning of the pressure plateau from 40 to 50 psi defines the $x/c$ position of separation over the bump for the various tunnel pressures. This also moves aft as the pressure increases since the interaction region becomes larger and the region itself moves aft with the shock wave.

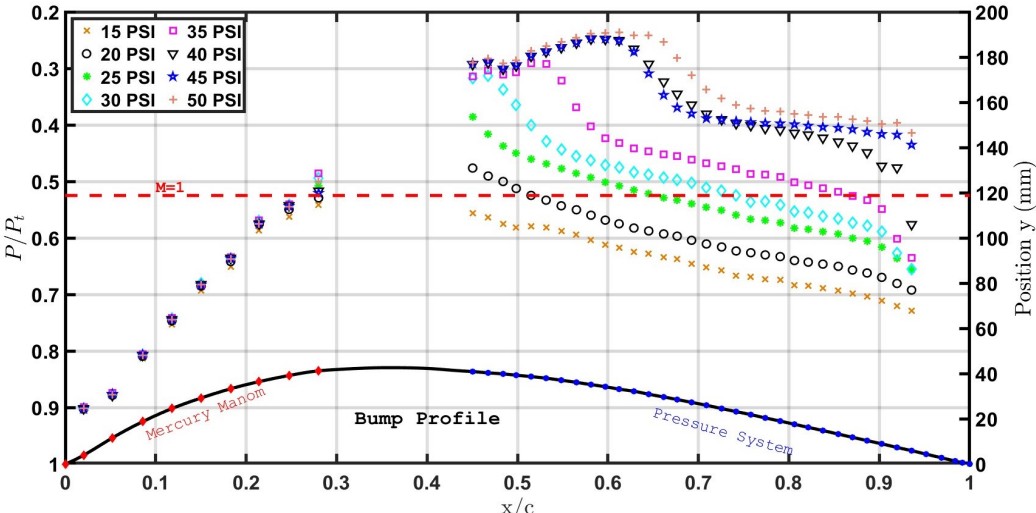

**Figure 7.** Evolution of the measured pressure ratio distribution for the different ejector blowing pressures.

## 3. Results

*Effects of Roughness on Surface $Cp$*

The pressure distribution on the bump gives information about the behaviour of the flow, and thus the effects of roughness. Applying the roughness, two major effects are expected, the shock bifurcation that can be identified by the change in length of interaction and the final pressure recovery. Figures 8–13 show the effect of the roughness strips on the pressure coefficient distribution in comparison with the smooth surface, when they are aligned with ($\phi = 0°$) the flow and spaced at different separation distances (5 mm and 20 mm). Data for the 10 mm separation cases are omitted for clarity.

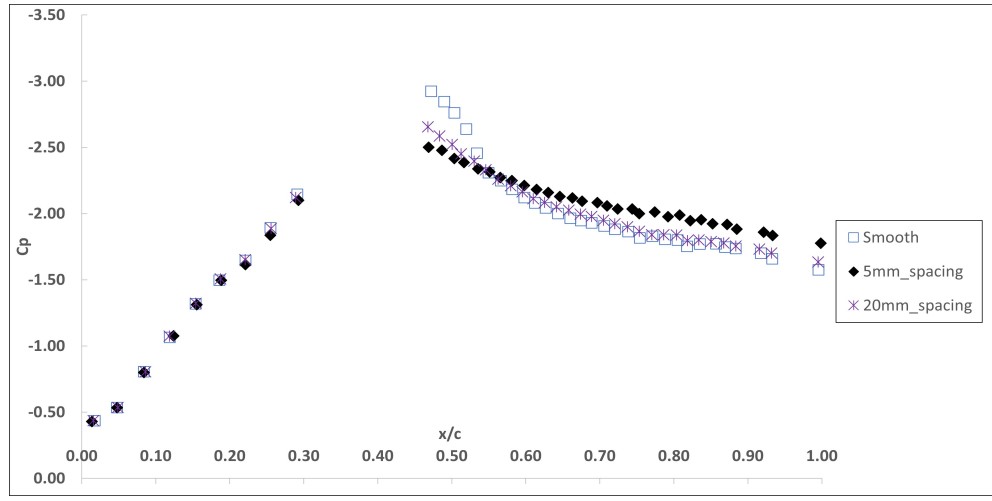

**Figure 8.** $Cp$ vs. $x/c$: Effect of roughness strips for 25 psi ejector blowing ($\phi = 0°$).

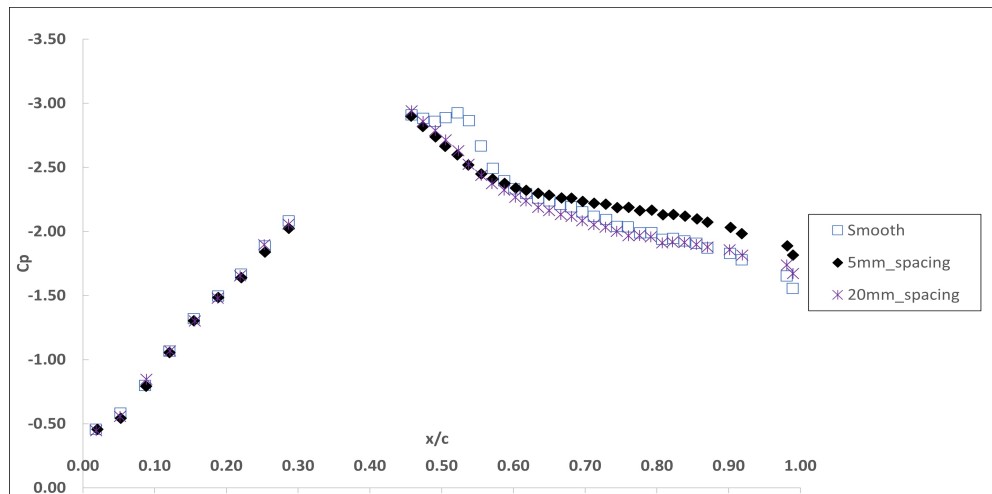

**Figure 9.** $Cp$ vs. $x/c$: Effect of roughness strips for 30 psi ejector blowing ($\phi = 0°$).

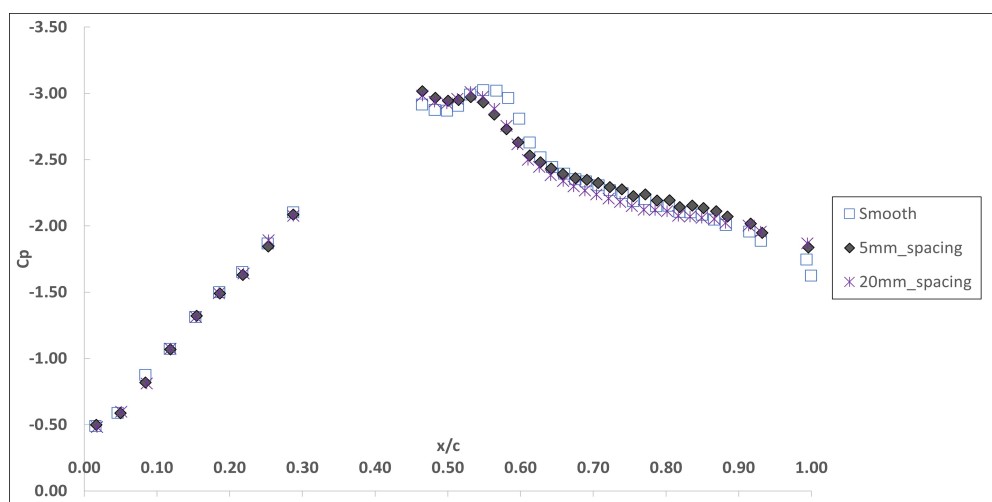

**Figure 10.** $Cp$ vs. $x/c$: Effect of roughness strips for 35 psi ejector blowing ($\phi = 0°$).

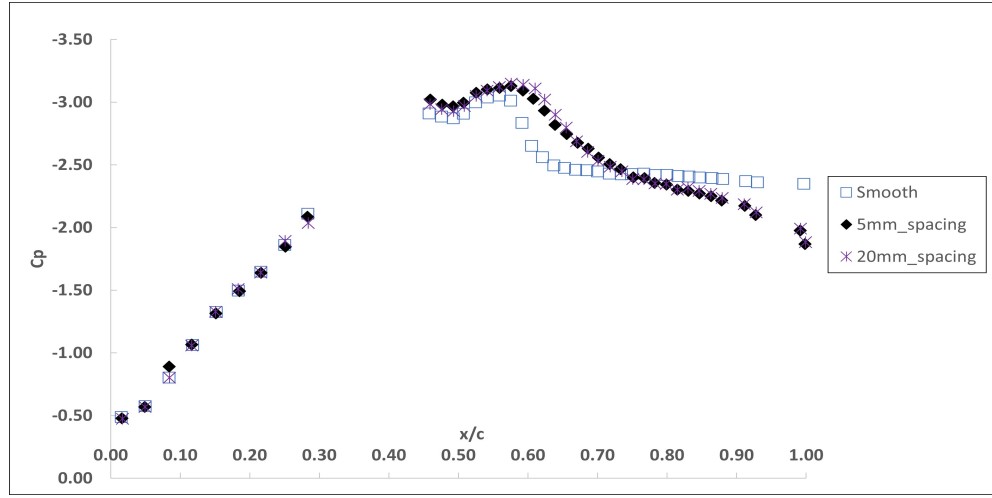

**Figure 11.** $Cp$ vs. $x/c$: Effect of roughness strips for 40 psi ejector blowing ($\phi = 0°$).

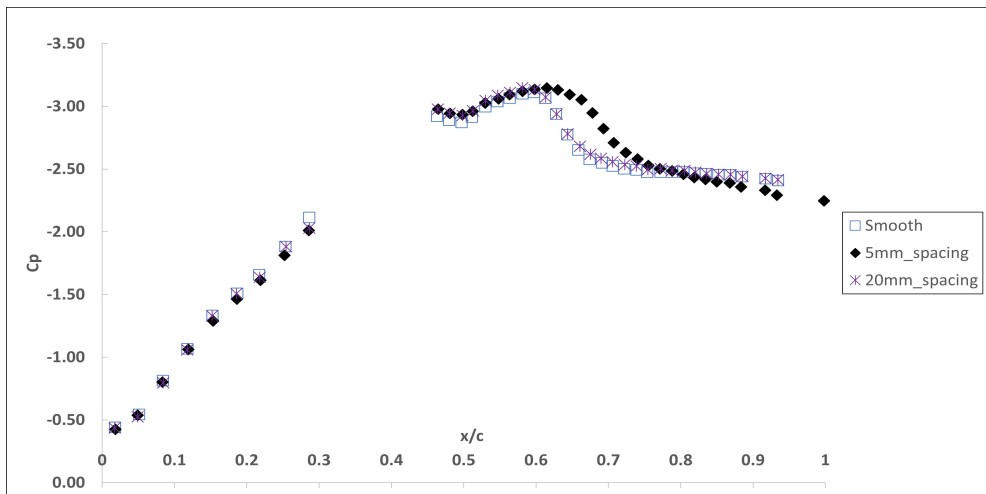

**Figure 12.** $Cp$ vs. $x/c$: Effect of roughness strips for 45 psi ejector blowing ($\phi = 0°$).

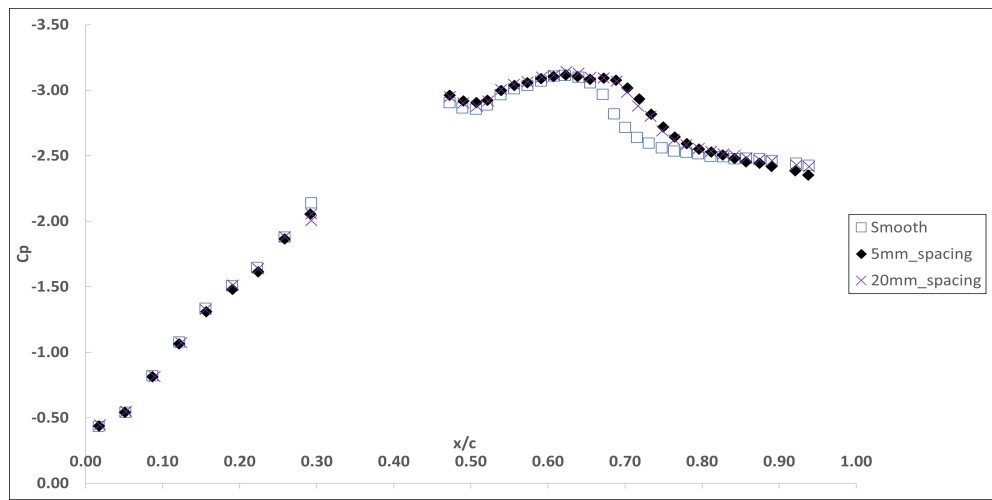

**Figure 13.** $Cp$ vs. $x/c$: Effect of roughness strips for 50 psi ejector blowing ($\phi = 0°$).

From these figures it is evident that between 25 and 40 psi ejector blowing, the 5 mm and 20 mm roughness configurations successfully move the shock wave further aft and improve the adverse pressure gradient. At 35 psi tunnel blowing pressure for both 5 mm and 20 mm spaced strips, the SWBLI region increased with a similar final $Cp$ value. At 40 psi blowing the shock bifurcation was considerably extended and the flow remain attached giving higher-pressure recovery at the rear of the bump in comparison to the smooth surface case. Shock bifurcation was again visible at 45 and 50 psi blowing pressure but more dominant for the 5 mm roughness case. Additionally, it was seen that the densest roughness configuration, i.e., 5 mm separation, performed better in comparison to the 20 mm one, as can be seen in Figure 14, where the shock location is plotted versus the ejector blowing pressure for the different strips spacings.

Specifically, for the high-density roughness (5 mm spaced) at 25 psi and 30 psi tunnel blowing pressure the smooth case exhibited a weak shock that was eliminated by adding the roughness. Adding the roughness at 40 psi shifted the shock further aft and suppressed separation, as also seen from the pair of shadowgraph images taken during the experiments, see Figure 15.

At higher tunnel pressures the effect is similar but less pronounced.

Figures 16–21 present the effect of the distributed roughness on the pressure coefficient distribution in comparison to the smooth surface, when they are skewed at an angle ($\phi = 30$)

to the flow and spaced at different distances (5 and 10 mm shown in this case) for increasing tunnel blowing pressures.

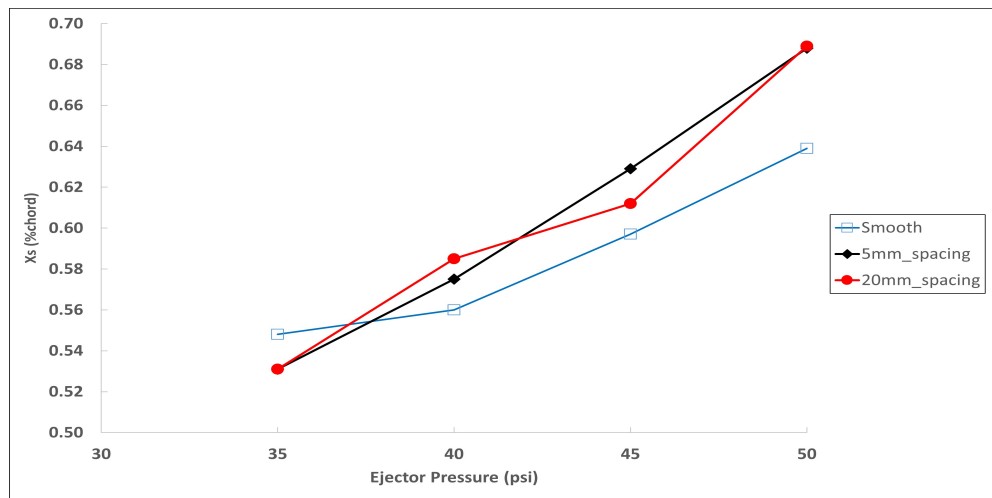

**Figure 14.** Shock location vs. ejector pressure.

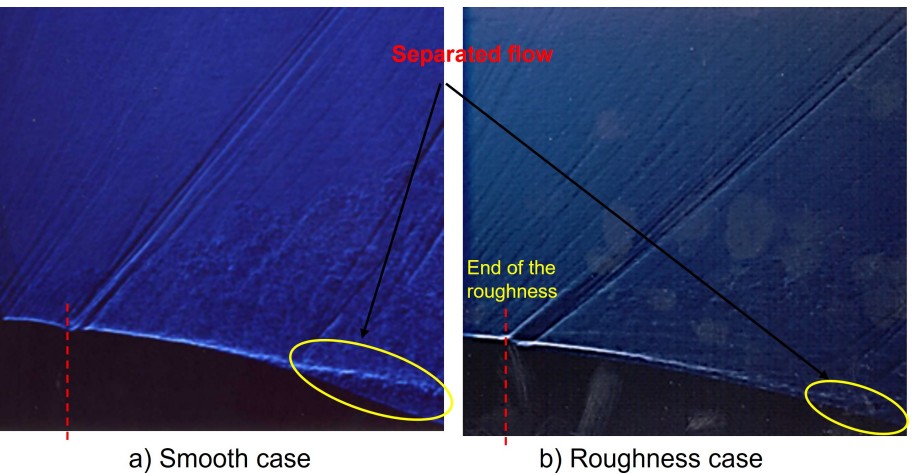

**Figure 15.** Shadowgraph image: Smooth vs. roughness (5 mm) effect at 40 psi. (0° deg).

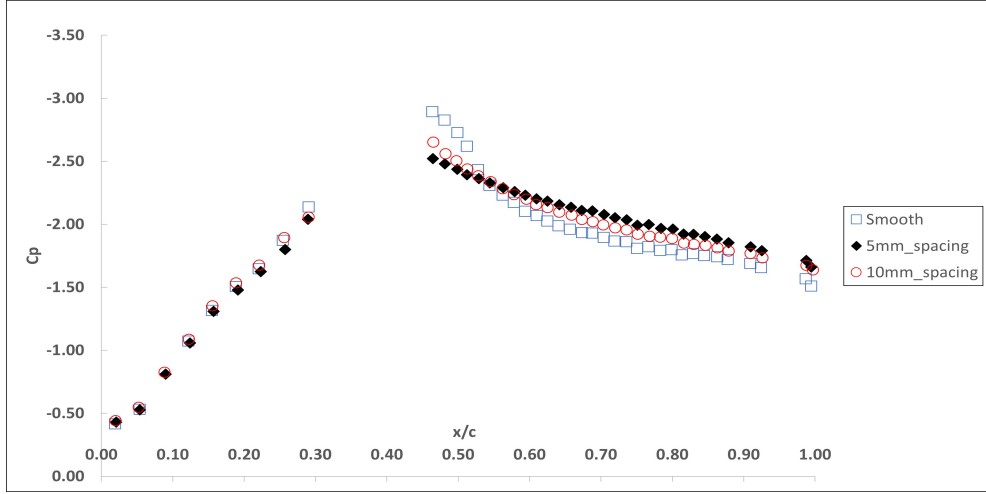

**Figure 16.** $Cp$ vs. $x/c$: Effect of roughness strips for 25 psi ejector blowing ($\phi = 30°$).

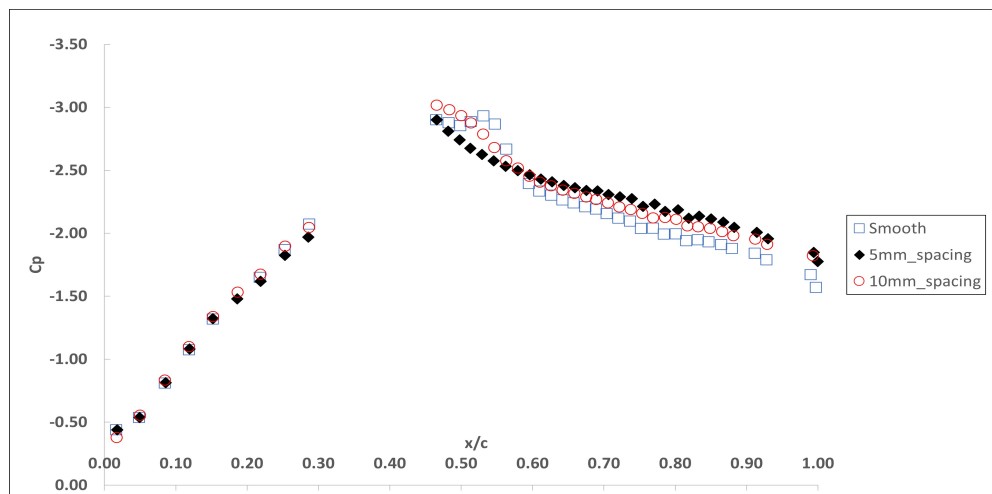

**Figure 17.** $Cp$ vs. $x/c$: Effect of roughness strips for 30 psi ejector blowing ($\phi$ = 30°).

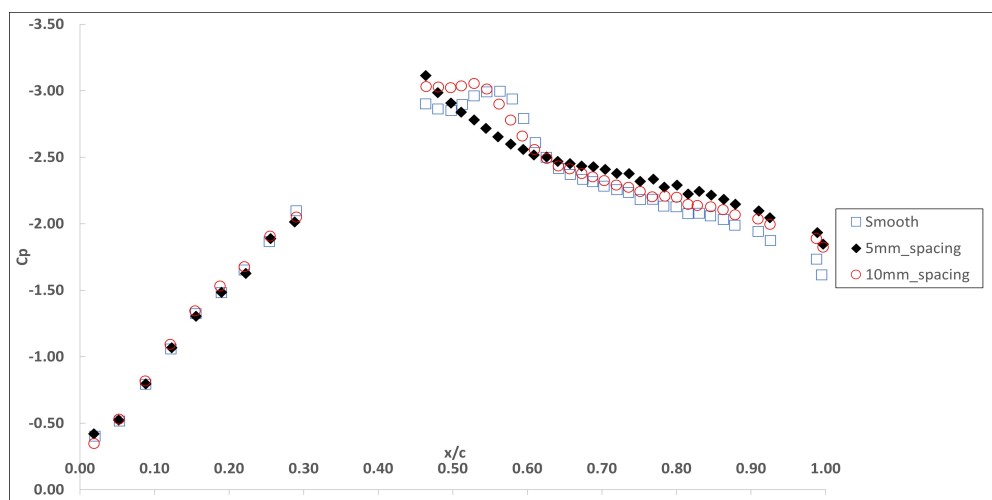

**Figure 18.** $Cp$ vs. $x/c$: Effect of roughness strips for 35 psi ejector blowing ($\phi$ = 30°).

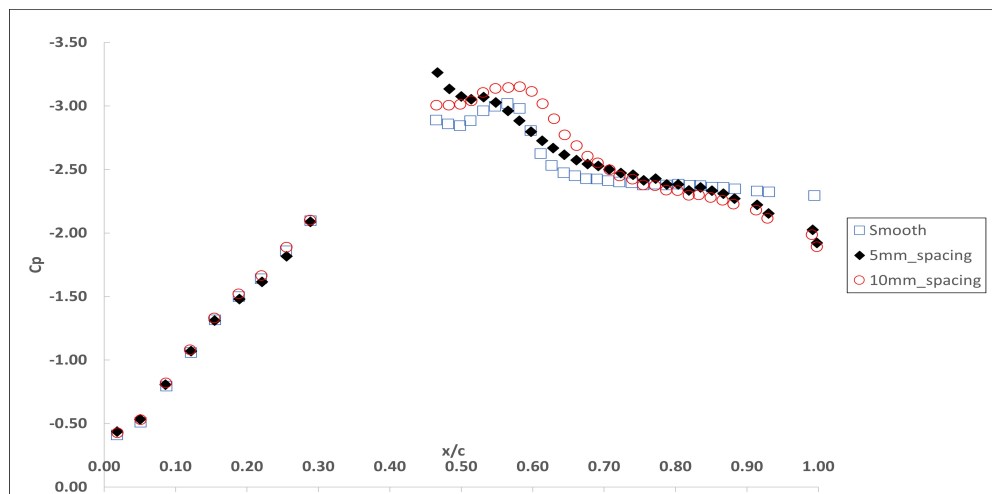

**Figure 19.** $Cp$ vs. $x/c$: Effect of roughness strips for 40 psi ejector blowing ($\phi$ = 30°).

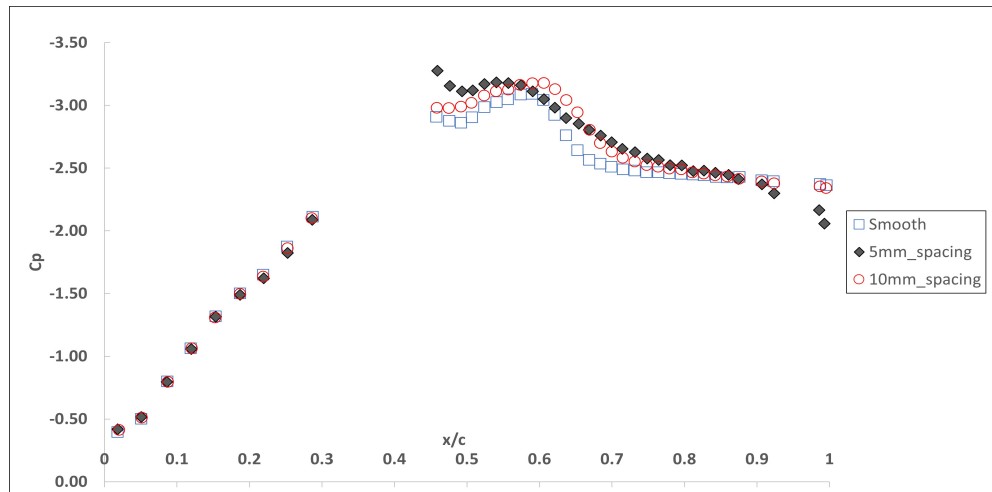

**Figure 20.** *Cp* vs. *x*/*c*: Effect of roughness strips for 45 psi ejector pressure.($\phi = 30°$).

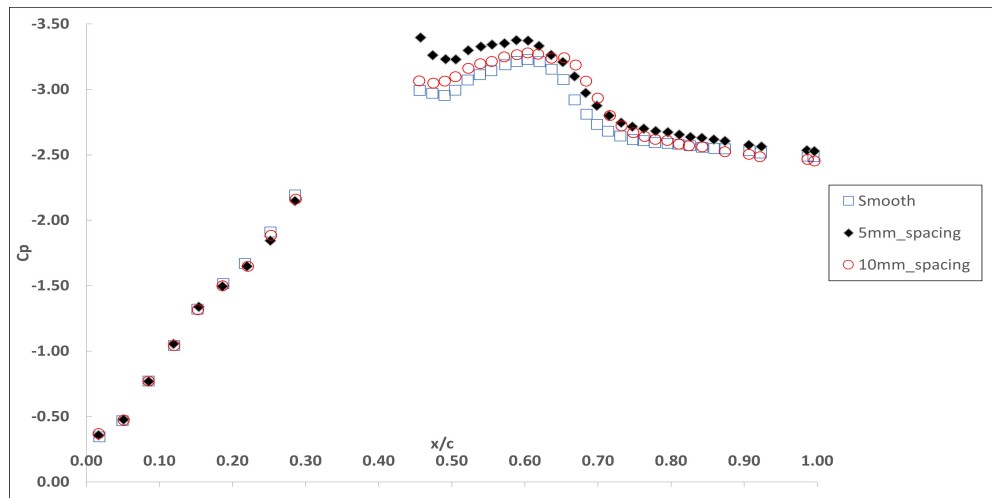

**Figure 21.** *Cp* vs. *x*/*c*: Effect of roughness strips for 50 psi ejector pressure ($\phi = 30°$).

The 5 mm 30° case gave the most promising results for the tunnel pressures tested. This configuration considerably bifurcated the shock wave at ejector pressures between 30 and 45 psi, and managed to keep the the flow attached for longer than the smooth and 10 mm cases.

Overall, the skewed roughness configuration was less promising when compared to the straight configuration at mid and higher ejector tunnel blowing pressures, but comparable for low tunnel blowing pressures (20–25 psi), as can be seen in Figure 22.

The results of this study, combined with all the surface pressure and shadowgraph flow visualisation images, showed that periodic strips of sub-boundary layer scale roughness can successfully bifurcate a transonic shock wave over the 14% thick bump model, thereby extending the shock interaction region, reducing the local adverse pressure gradient and suppressing shock-induced boundary layer separation. More research is needed to derive definite conclusions about the optimal roughness geometries. It is suggested that further experiments, involving more detailed measurements including boundary layer and shear layer measurements, the analysis of surface pressure data transients, and high-fidelity Navier–Stokes-based simulations with roughness modelling should be undertaken.

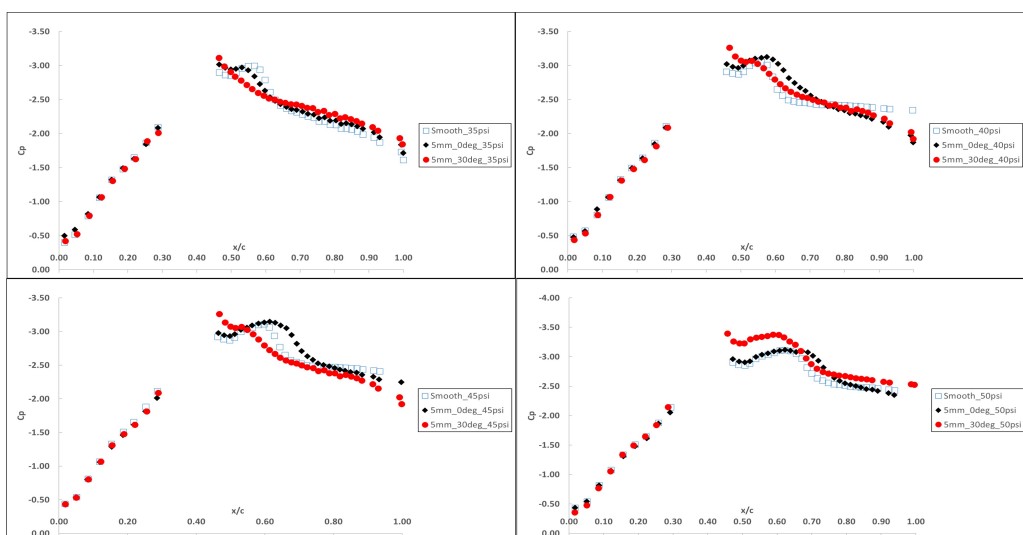

**Figure 22.** $Cp$ vs. $x/c$L Effect of high density roughness strips at different psi ejector pressure.

## 4. Conclusions

The conclusions derived from this study are:

(1) Periodic strips of sub-boundary layer scale roughness can successfully bifurcate a transonic shock wave over the 14% thick bump model, thereby extending the shock interaction region, reducing the local adverse pressure gradient and suppressing shock-induced boundary layer separation.

(2) The results show that strips aligned with the stream-wise flow with the minimum 5 mm separation distance (as opposed to 10 mm and 20 mm) gave the best shock bifurcation and separation suppression effect, akin to the effect of solid shock bumps.

**Author Contributions:** S.P. conceptualized and performed the experiment, D.D.P. performed data curation, post-processing and wrote the manuscript. All authors have read and agreed to the published version of the manuscript.

**Funding:** This research received no external funding.

**Institutional Review Board Statement:** Not applicable.

**Informed Consent Statement:** Not applicable.

**Data Availability Statement:** Some or all the data, models, or codes that support the findings of this study are available from the corresponding authors upon reasonable request.

**Conflicts of Interest:** The authors declare no conflicts of interest.

## Abbreviations

| | |
|---|---|
| $c$ | Bump chord length |
| $C_p$ | Pressure coefficient |
| $d$ | Distance between the strip |
| $\Delta_B$ | Shift of the velocity profile |
| $k_s$ | Equivalent sand grain roughness height |
| $k_s^+$ | Non-dimensional roughness height, $k_s u_\tau / \nu$ |
| M | Mach number |
| $\nu$ | Kinematic viscosity |
| P | Static pressure |
| $P_T$ | Total Pressure |
| $u^+$ | Non-dimensional velocity |
| $u_\tau$ | Friction or shear velocity |
| $y^+$ | Non-dimensional distance from the wall |

| | |
|---|---|
| $\phi$ | Angle of the strips |
| $\rho$ | Density |
| $\tau_w$ | Shear stress at the wall |
| $X_S$ | Shock position |

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
