# Peer review of "Passive Transonic Shock Control on Bump Flow for Wing Buffet Suppression"

_aerospace, doi:10.3390/aerospace10060569_

Round 1
Reviewer 1 Report
Congratulations to this interesting and thorough experimental investigation!
I have some smaller remarks:
Line 19: I am sure that "cheaper and faster" are no drivers of the fullfillment of the EU Flightpath 2050 vision and will reduce the environmental impact of air transport :-) Maybe you want to replace with "quieter"?
Line 31: "complicated" is a very unspecific and unphysical word. I am sure, you will find a better one.
Unfortunately all your figures are difficult to see: Symbols are difficult to distinuish (e.g. square and circle") as they are too small. Legends and axle markings could be larger.
Despite all this I have one major point that makes it impossible for me to agree to publish the article in its current form: the "ejector blowing pressure" you use as a variation (of what?) is no appropriate measure that lets others follow or reproduce your results. You must change to and use basic measures of flow physics like Mach-number, Reynolds-number, static or dynamic pressure. If others do not have the possibility to experimentally reproduce or numerically simulate your results, it does not correspond to a good scientific practice.
Overall the English language is very good. Except from line 218 there is a break and spelling errors accumulate:
Line 218: "the the"
Line 252: "Shock loccation"
Line 253: "teh diferent"
Line 258: "experiments ,in"
Line 265: "30° deg"
Line 267: "the the"
Line 45: "as"
Author Response
- "cheaper and faster" changed with "cleaner and quieter".
- "complicated" changed with "complex".
- All the figures were improved making the symbols, legends, and axis marker bigger.
- the "ejector blowing pressure" is related to the wind tunnel outlet static pressure and a graph shown this relation was added in the paper.
- Minors grammar mistakes were fixed
Reviewer 2 Report
The article addresses the shock wave/boundary layer interaction in transonic airflow over a convex wall. The aim of work is to study experimentally an effect of a small roughness of the wall on the boundary layer thickness and dissipation of the shock in a vicinity of its foot. The obtained results are of significant interest. A few comments on the text are as follows:
1) In lines 164-165 the authors point out that “a strong normal shock wave was generated” in the baseline flow. In fact, the shock is not strictly normal; moreover, the flow in the small test section is essentially different from the one over an aerofoil sketched in Fig. 1. That is why, it would be relevant to clearly describe the shape of local supersonic zone over the bump and/or provide a shadowgraph image of the full shock up to its top.
2) The long part of text from line 19 to line 100 should be entitled as a first subsection of Section 1.
3) Directions respective to “0.2 m x 0.25 m x 0.5 m” need to be pointed out in line 170.
4) Throughout the manuscript, physical quantities must be typed in italic.
1) The authors should choose between “shock-wave” and “shock wave” (see, e.g., line 226) and avoid using both versions randomly in the manuscript.
2) Incorrect expressions x /c 0.47 and x /c 0.63 in lines 227, 228 must be replaced by x /c = 0.47 and x /c = 0.63.
3) There are misprints in the text, for example, “loccation” in line 252, “teh” in line 253, “Seperated” in line 327.
4) In the list of references, about 50% of titles are typed using capital characters for the beginnings of words, whereas other 50% are typed with lowercase characters.
Author Response
- The shape of local supersonic zone over the bump and a shadowgraph image was added in the paper.
- The long part of the text from line 19 to 100 was entitled "background" as subsection.
- Direction indicated: which has a rectangular test section size of 0.2 m (height) x 0.255 m (width) x 0.5 m (length)”.
- All physical quantities typed in italic.
Quality of English Language
- Throughout the manuscript “shock wave” was used.
- Incorrect expressions fixed.
- mistyping fixed.
- All the references have the same format
Round 2
Reviewer 1 Report
Thank you very much for re-working and improving the paper!